# Modeling Degraded Bamboo Shoots in Southeast China

**Xiao Zhou [1,2], Fengying Guan [1,2,*], Shaohui Fan [1], Zixu Yin [1,2], Xuan Zhang [1,2], Chengji Li [1,2] and Yang Zhou [1,2]**

[1] International Center for Bamboo and Rattan, Key Laboratory of National Forestry and Grassland Administration, Beijing 100102, China

[2] National Location Observation and Research Station of the Bamboo Forest Ecosystem in Yixing, National Forestry and Grassland Administration, Yixing 214200, China

* Correspondence: guanfy@icbr.ac.cn; Tel.: +86-10-84789808

**Abstract:** Degraded bamboo shoots (DBS) constitute an important variable in the carbon fixation of bamboo forests. DBS are useful for informed decision making in bamboo forests. Despite their importance, studies on DBS are limited. In this study, we aimed to develop models to describe DBS variations. By using DBS data from 64 plots of Yixing forest farm in Jiangsu Province, China, a mixed-effects model was constructed, including block-level random effects. We evaluated the potential impact of several variables on DBS. The number of bamboo shoots (NBS), mean height to crown base (MHCB), hydrolytic nitrogen (HN), and available potassium (AK) significantly contributed to the model. By introducing the block-level random effect in the logistic model, the fitting statistics were significantly improved. The model showed that there were increased DBS in bamboo stands with decreased MHCB and AK, whereas DBS decreased with decreasing NBS and HN. The application of K fertilizer reduced the number of DBS during the emergence stage. By adjusting these factors, the number of DBS in bamboo forests can be reduced, which provides a theoretical basis for increasing the biomass of bamboo forests. It can also provide an important basis for studying the carbon sink characteristics of bamboo forests and help to formulate more effective bamboo forest management plans.

**Keywords:** basic model; mixed-effects model; model validation; bamboo management

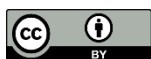

## 1. Introduction

Bamboo forests are important forest type. Moso bamboo (*Phyllostachys edulis*) is mainly distributed in tropical and subtropical regions [1,2]. It has economic (as food and wood) and ecological benefits [3,4]. Degraded bamboo shoots (DBS) are an important characteristic of bamboo forest dynamics. DBS are a phenomenon in which the development of new bamboo stops after the shoots are unearthed because of insufficient nutrition in the soil, foreign pests and diseases, sudden cold or dry weather, competition in the bamboo forest (such as the number of bamboo shoots, NBS), and bamboo forest structure, and they then die and cannot further develop into bamboo. They are thought to play a role in competition and regulation of bamboo forests [5].

DBS are described as follows: (1) DBS taper to a large degree; the color of the shoot sheath is dark, dark brown, or even black; and the sheath leaves are underdeveloped; (2) the surface fur on the middle and upper bamboo shoot sheaths is disordered, and the shoot is easily damaged and can fall off when touched; there is no white powder or obvious growth in the sheath; (3) when pinching the bamboo shoot tip with your hand, it is hard and has no obvious elasticity; and (4) when the bamboo shoot sheath is peeled off, the color of the lower sheath is purple, often with cyan blue stripes, or cyan to dark yellow; the bamboo shoots are dark yellow, and the color of the root tip changes from purplish red to cyan to yellow (Figure 1).

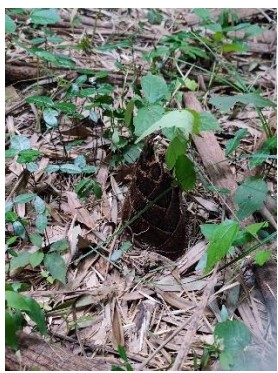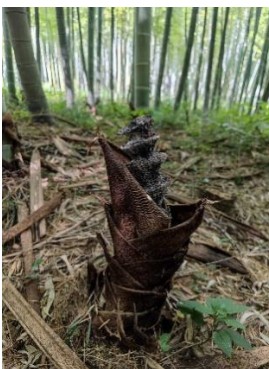

**Figure 1.** Phenomenon of degraded bamboo shoots in a bamboo forest.

The proportion of DBS is as high as 80% in some bamboo species, which substantially reduces the yield of shoots and culms and can seriously threaten the sustainability of bamboo forests [6–8]. In recent years, large-scale DBS have been reported in bamboo forests in bamboo-growing regions. Therefore, it is important to understand the impact of different sites and environments on DBS [6,8].

In bamboo forests, bamboo shoots can be divided into normal and abnormal DBS. Normal DBS are mainly caused by the competition of water, nutrients, and light conditions in new bamboo. Abnormal DBS are mainly caused by man-made random interference or natural disasters, such as extreme weather, forest fires, snow disasters, droughts, forest diseases, and insect pests [4,5]. Owing to the randomness of abnormal DBS, only normal DBS stands are typically studied. Therefore, we considered only the effects of nutrient content and stand variables on DBS. DBS consume a considerable amount of nutrients from the mother bamboo, but this does not increase the yield, which could lead to a reduction in NBS and the production potential of bamboo stands. This can increase economic losses caused by the return of shoots to the ground [5,8]. Thus, an accurate description of DBS conditions provides a theoretical basis for effective maintenance and use of bamboo shoots.

DBS, a key feature in the growth and harvest of bamboo forests, are important for characterizing the forest structure and composition [5,9]. As mature bamboo can be harvested in 4–6 years, an increase in the number of DBS may have a negative impact on the structural stability and ecological benefits of bamboo forests, affecting forest mortality and impairing sustainable management of the forest [10]. Compared with other stand characteristics, it is more challenging to accurately describe the pattern of DBS in a given site condition, as DBS have high variability in space and time [7,10].

Previous studies have focused more on the development stages of DBS [11–13], the types and causes of bamboo degradation [14,15], and the general features of DBS [5,16,17] rather than accurately modeling DBS. Only a few researchers have attempted to study normal bamboo shoots and DBS modeling [18,19].

DBS are the result of several factors. The degradation process of bamboo shoots is complex, with multifactor synergism and a considerable degree of randomness; therefore, the underlying mechanisms remain elusive [20], limiting its modeling capability.

DBS may vary among different bamboo stand structures. DBS data were obtained from bamboo forests with different sites and environments. Therefore, the data were hierarchically structured (a sample plot nested in the blocks), and observations are most likely to be spatially correlated. Mixed-effects modeling can effectively solve these problems [21,22]. In addition, the method considers randomness in the data and potential variables caused by randomness, thus improving the accuracy of the model [21–24]. To describe variations in DBS, similar model types that describe forest mortality [4,25,26] and forest fires [27–29] are necessary. These model types include Poisson, negative binomial, zero-inflated, and hurdle models [23,24], among others [4,25].

It is assumed that soil nutrient content and stand variables can affect the competitive relationship between moso bamboo individuals, provide a suitable environment for moso bamboo growth, and promote growth and development, thus impacting DBS. To resolve the above-mentioned issues (the effects of soil nutrient content and stand variables, hierarchically structured data, and spatial correlations), we aimed to (1) develop a model describing variations in degraded bamboo shoots through the application of mixed-effects modeling, (2) evaluate the sensitivity of the DBS model to soil nutrient content, and (3) evaluate the effects of stand factors and soil nutrients on bamboo shoot degradation. By adjusting these variables, the models can provide technical support for the increase in carbon reserves in bamboo forests and help formulate more effective bamboo forest management plans. The presented models provide a theoretical basis for estimating bamboo forest growth in southeast China to maintain bamboo shoots and make full use of degraded shoots.

## 2. Materials and Methods

### 2.1. Study Site

The experimental sample plots were set in Yixing, Jiangsu Province (31°15′1″–31°15′40″ N, 119°43′52″–119°44′41″ E). The average precipitation is 1167 mm, the temperature is 15.7 °C, and the annual evaporation is 886.8 mm. The area is mainly low mountains and hills and the soil type is yellow clay (GB/T 17296-2009). The vegetation type is moso bamboo forest.

The traditional manual management measures adopted in the moso bamboo forest include bamboo cutting, shrub cutting, grass cutting, mining winter and spring bamboo shoots, and tourism. All human activities were controlled in the study area.

We established 64 temporary sample plots (TSPs) on a Yixing farm, and each sample plot was 20 × 3 m (Figure 2). Because the slope, aspect, and slope position may lead to different growth environments, we divided the sample plots into different blocks. The sample plots are listed in Table 1.

**Table 1.** Block division and plot setting.

| Slope (°) | Aspect | Slope Position | Number of Sampling Plots | Block |
|---|---|---|---|---|
| 0–3 | Southwest | Downhill slope | 25 | Block 1 |
| 4–7 | Southwest | Middle slope | 17 | Block 2 |
| 8–11 | Southwest | Uphill slope | 12 | Block 3 |
| 11–14 | Northwest | Middle slope | 5 | Block 4 |
| 15–18 | Northwest | Middle slope | 5 | Block 5 |

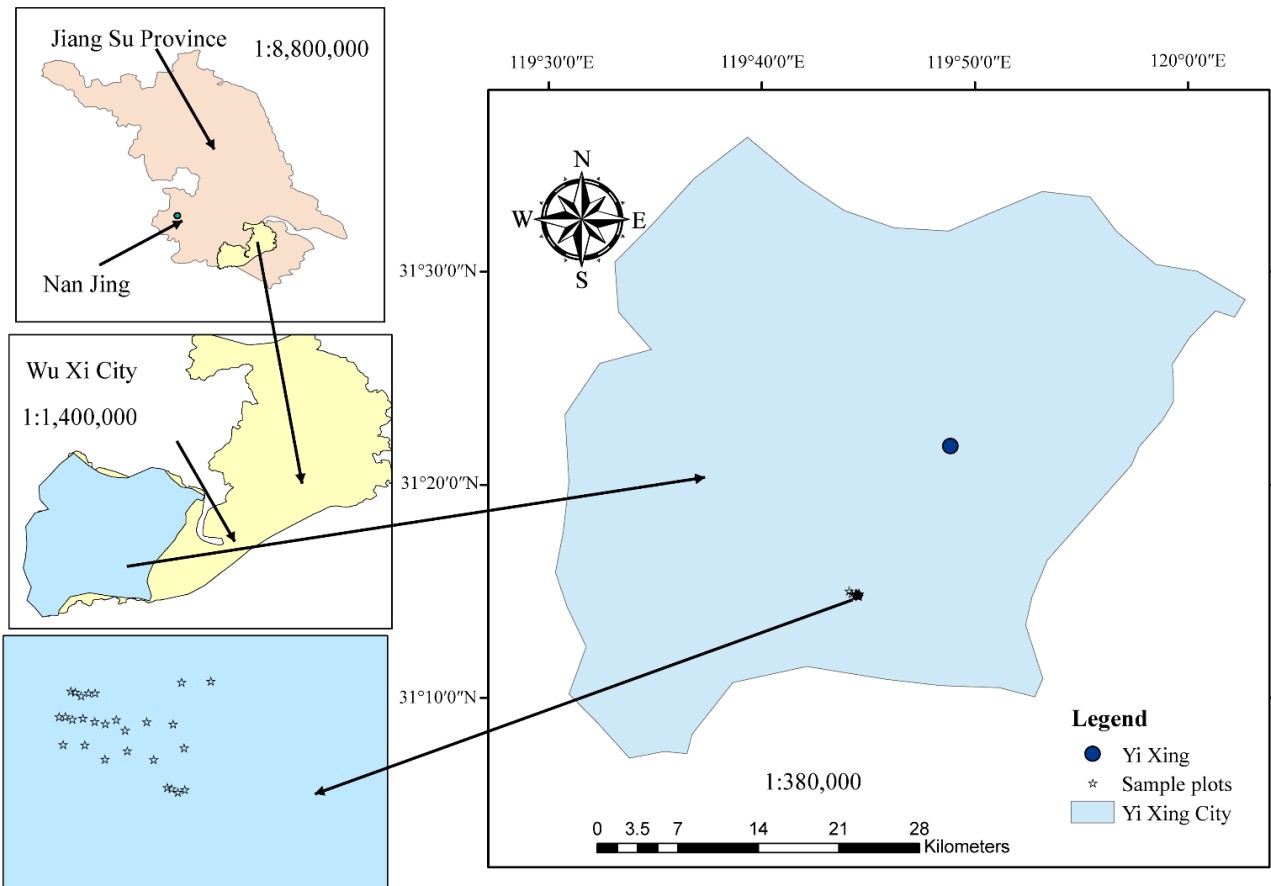

**Figure 2.** Study area showing the location of the sample plots.

### 2.2. Sampling and Measurement of DBS

#### 2.2.1. Distinguishing Degraded Bamboo Shoots (DBS)

Moso bamboo DBS can mainly be distinguished using four characteristics: (1) from a distance, the tapering degree of the retreating shoot is large; the color of the shoot sheath is dark, dark brown, or even black; and the sheath leaves are underdeveloped; (2) when looking at the degradation of bamboo shoots, the surface fur on the middle and upper bamboo shoot sheaths is disordered, it is not tall and straight or dry, and most of the hair is dry; it is easily damaged and can fall off when touched; and there is no white powder or obvious growth in the sheath; (3) when pinching the bamboo shoot tip with your hand, it is hard and has no obvious elasticity; and (4) when the bamboo shoot sheath is peeled off, the color of the lower sheath is purple, often with cyan blue stripes, or cyan to dark yellow; the bamboo shoots are dark yellow, and the color of the root tip changes from purplish red to cyan to yellow [18].

#### 2.2.2. Degraded Bamboo Shoots (DBS) Investigation

Using these characteristics, we selected sample plots of different groups in the Yixing state-owned forest farm to investigate the DBS. The investigation period was from March to May 2019 and the investigation event lasted for 50 days. Bamboo shoots unearthed in the sample plot were observed daily to determine whether they survived. The TSPs cover a wide range of bamboo forests with varying stand structures, stand densities, bamboo sizes, bamboo ages, site productivity, terrain, and environment. All diameters at breast height (DBH ≥ 5) were measured to obtain the DBH, height to crown base (HCB), and number of bamboo shoots (NBS). We used the direct counting method to obtain the NB, NBS, and DBS. Subsequently, various measures describing bamboo stand density were

derived from these measurements. The distribution patterns of the DBS are shown in Figure 3. The bamboo stand variables are presented in Table 2.

**Table 2.** Summary statistics of measurements of bamboo variables and soil nutrient content. DBS, degraded bamboo shoots; NBS, number of bamboo shoots; MD, mean diameter at breast height; MHCB, mean height to crown base; N, stand density of bamboo forest; NB, number of new bamboo; BA, basal area of all bamboo; QMD, quadratic mean DBH; SOC, soil organic carbon; TN, total nitrogen; TP, total phosphorus; TK, total potassium; HN, hydrolytic nitrogen; AP, available phosphorus; AK, available potassium.

| Variables | Min | Max | Mean | Std. |
|---|---|---|---|---|
| NBS (culms/ha) | 1000 | 4867 | 2373 | 901.75 |
| MD (cm) | 7.42 | 9.82 | 8.97 | 0.42 |
| MHCB (m) | 3.41 | 5.98 | 4.53 | 0.53 |
| N (culms/ha) | 1833 | 8100 | 4507 | 1271.17 |
| DBS (culms/ha) | 0 | 2700 | 611 | 620.20 |
| NB (culms/ha) | 813 | 3400 | 1762 | 688.92 |
| BA (m²/ha) | 13.15 | 49.56 | 29.14 | 7.76 |
| QMD (cm) | 7.64 | 9.91 | 9.10 | 0.40 |
| SOC (g/kg) | 13.44 | 43.10 | 33.69 | 5.47 |
| TN (g/kg) | 0.84 | 2.34 | 1.74 | 0.27 |
| TP (g/kg) | 0.22 | 0.39 | 0.26 | 0.04 |
| TK (g/kg) | 7.84 | 11.76 | 9.63 | 0.77 |
| HN (mg/kg) | 104.9 | 221.10 | 164.6 | 21.65 |
| AP (mg/kg) | 0.17 | 2.45 | 1.17 | 0.48 |
| AK (mg/kg) | 46.34 | 84.24 | 60.45 | 7.61 |

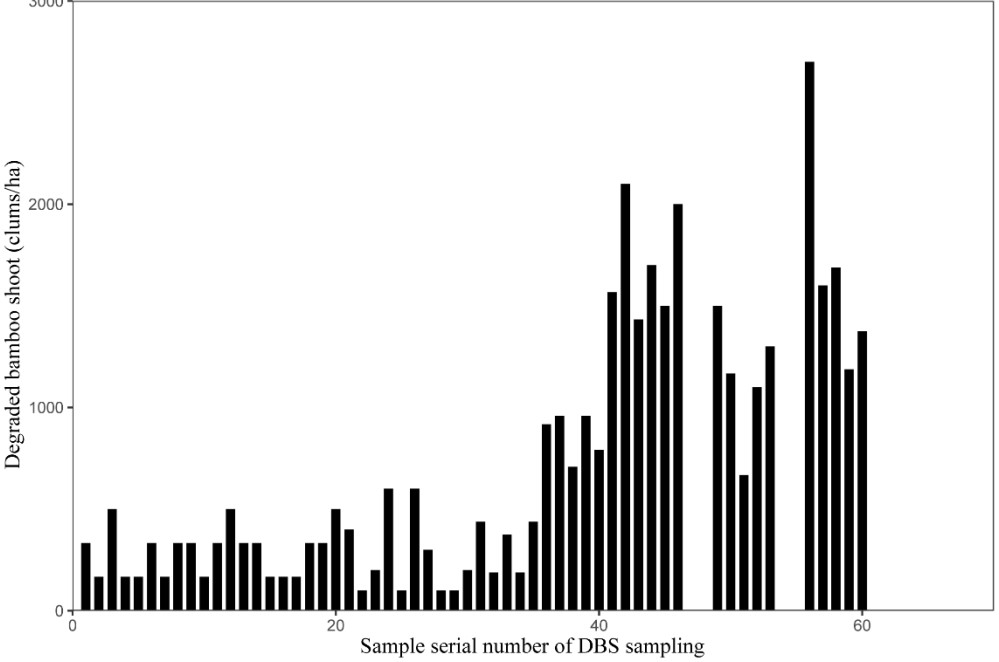

**Figure 3.** Distribution patterns of degraded bamboo shoots of moso bamboo.

2.2.3. Sample Plots Soil Sampling

Due to the apparent aggregation of soil nutrients, studies have found that the nutrient-absorbing fine roots of moso bamboo in this area are mainly distributed in the soil surface (0–10 cm) [30]. Therefore, we mainly focused on the contribution of soil nutrients in the 0–10 cm layer.

In March 2019, soil samples of 0–10 cm soil layer were collected from 64 TSPs (five samples per sample plot). The soil samples were mixed in a sample plot to form a composite sample and brought to the laboratory. In total, 64 samples were collected.

### 2.2.4. Determination of Soil Nutrients

The soil samples were passed through 0.25 mm and 0.15 mm sieves to determine the total nitrogen, total phosphorus, and total potassium contents. Additionally, the samples were passed through a 1 mm sieve for the determination of alkali hydrolyzable nitrogen, available phosphorus, and available potassium.

The soil organic matter content was determined using the potassium dichromate oxidation external heating method. See Zheng et al. for the determination method of soil nutrient content [31]. Table 2 shows the statistics of soil nutrient content.

### 2.3. Determination of Predictor Variables

We evaluated bamboo size, competition, and nutrient content, which significantly influence DBS. Predictor variables affecting DBS were selected based on their biological significance and logic. Fifteen variables were evaluated for their potential influence on DBS. Bamboo individual and stand variables were used, including MD, MHCB, NBS, N, and NB. The competition variables used in this study were the BA and QMD. Nutrient contents included SOC, TN, TP, TK, HN, AP, and AK. We explained the site by using block-level variables as random effects.

Although many variables were evaluated, only a few variables that have significant impact on DBS and are not significantly related to each other were selected. We used the variance inflation factor (VIF) to control for potential collinearity effects between predictor variables [32]. Collinearity produces large standard errors in the parameter estimates (VIF > 2), which causes a large bias [33]. Therefore, only the following predictor variables with a VIF < 2 were retained in our final model: NBS, MHCB, AK, and HN.

### 2.4. Candidate Models

We used seven versatile mathematical functions, hereafter referred to as the basic models, which are commonly used for count data modeling (variables of the presence or absence) to develop models for describing variations in degraded bamboo shoots (Table 3). These models are the Poisson model (PS) and negative binomial model (NB), which refer to the standard function, zero-inflated Poisson model (ZIP), zero-inflated negative binomial model (ZINB), hurdle Poisson model (HP), hurdle negative binomial model (HNB), and logistic regression model (Equations (1)–(7)). First, the basic models were fitted using NBS, MHCB, AK, and HN as predictors, and their fitting performance was compared. Second, we selected the model with the best fit to formulate a mixed-effects model with the inclusion of block-level random effects.

**Table 3.** Summarized forms of DBS functions, which were expanded through the inclusion of four predictor variables.

| Model | Equation | Equation No. |
|---|---|---|
| PS | $\log(\lambda_{ij}) = \log(DBS_{ij}) = \beta_0 + \beta_1 NBS_{ij} + \beta_2 MHCB_{ijk} + \beta_3 AK_{ijk} + \beta_4 HN_{ij}$ | (1) |
| NB | $\log(\lambda_{ij}) = \log(DBS_{ij} + e_{ij}) = e_{ij}(\beta_0 + \beta_1 NBS_{ij} + \beta_2 MHCB_{ijk} + \beta_3 AK_{ijk} + \beta_4 HN_{ij})$ $Exp(e_{ij}) \sim Gamma(\theta^{-1}, \theta^{-1})$ | (2) |
| ZIP | $\begin{cases} \log(\dfrac{p_{ij}}{1-p_{ij}}) = \alpha_0 + \alpha_1 AK_{ij} \\ \log(\lambda_{ij}) = \log(DBS_{ij} / (1-p_{ij})) = \beta_0 + \beta_1 NBS_{ij} + \beta_2 MHCB_{ijk} + \beta_3 AK_{ijk} + \beta_4 HN_{ij} \end{cases}$ | (3) |

| ZINB | $\begin{cases} \log(\dfrac{p_{ij}}{1-p_{ij}}) = \alpha_0 + \alpha_1 AK_{ij} \\ \log(\lambda_{ij}) = \log((DBS_{ij} + e_{ij})/(1-p_{ij})) = \beta_0 + \beta_1 NBS_{ij} + \beta_2 MHCB_{ijk} + \beta_3 AK_{ijk} + \beta_4 HN_{ij} \\ Exp(e_{ij}) \sim Gamma(\theta^{-1}, \theta^{-1}) \end{cases}$ | (4) |
|---|---|---|
| HP | $\begin{cases} \log(\dfrac{p_{ij}}{1-p_{ij}}) = \alpha_0 + \alpha_1 AK_{ij} \\ \log(\lambda_{ij}) = \beta_0 + \beta_1 NBS_{ij} + \beta_2 MHCB_{ijk} + \beta_3 AK_{ijk} + \beta_4 HN_{ij} \\ DBS_{ij} = (1-p_{ij})\lambda_{ij}/(1-e^{-\lambda_{ij}}) \end{cases}$ | (5) |
| HNB | $\begin{cases} \log(\dfrac{p_{ij}}{1-p_{ij}}) = \alpha_0 + \alpha_1 AK_{ij} \\ \log(\lambda_{ij}) = \log((DBS_{ij} + e_{ij})/(1-p_{ij})) = \beta_0 + \beta_1 NBS_{ij} + \beta_2 MHCB_{ijk} + \beta_3 AK_{ijk} + \beta_4 HN_{ij} \\ Exp(e_{ij}) \sim Gamma(\theta^{-1}, \theta^{-1}) \end{cases}$ | (6) |
| Logistic | $DBS_{ij} = \dfrac{a}{1 + e^{(\beta_0 + \beta_1 NBS_{ij} + \beta_2 MHCB_{ijk} + \beta_3 AK_{ijk} + \beta_4 HN_{ij})}}$ | (7) |

Notes: $DBS_{ij}$ is the number of degraded bamboo shoots in the $j$th sample plot nested in the $i$th block; $NBS_{ij}$ is the number of bamboo shoots in the $j$th sample plot nested in the $i$th block; $MHCB_{ij}$ is the mean height to crown base in the $j$th sample plot nested in the $i$th block; $AK_{ij}$ is the available phosphorus content in the 0–10 cm soil layer of the $j$th sample plot nested in the $i$th block; $HN_{ij}$ is the hydrolytic nitrogen content in the 0–10 cm soil layer of the $j$th sample plot nested in the $i$th block; $a, \beta_0 - \beta_4$ are estimated parameters; $\lambda_{ij}$ represents the mean number of counts in a given period within the expected value of the model. See Zhou et al. 2021 for candidate model details [23,24].

### 2.5. Mixed-Effects Models

Bamboo shoot data were collected from different bamboo forests growing under different growth conditions to simulate the relationship between degraded bamboo shoots, stands, and soil content variables. Therefore, the data were hierarchically structured (investigation of different sample plots in the same block).

By introducing block-level random effects, seven basic models were used to develop a one-level nonlinear mixed-effects (NLME) DBS model. We considered the combination of all parameters. Models with the smallest Akaike information criterion (AIC) and largest log-likelihood (LL) were selected for further analysis. Figure 4 shows the distribution of different predictors and DBS.

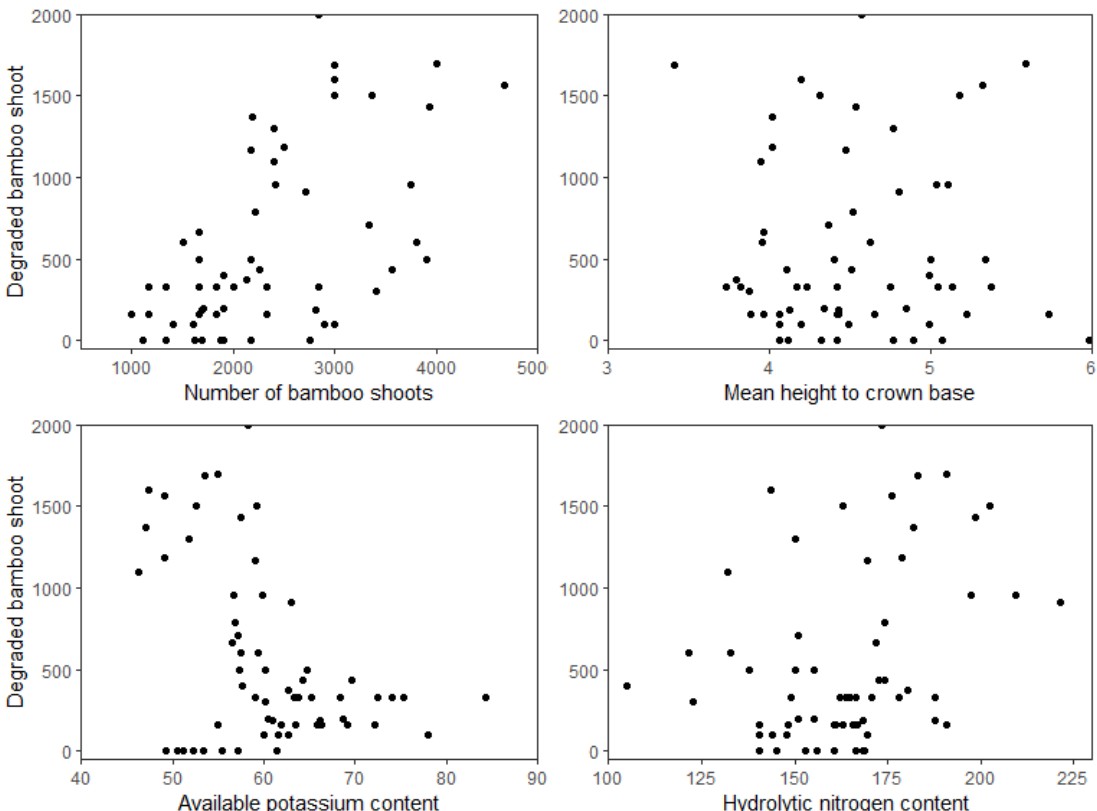

**Figure 4.** Scatter plot distribution between degraded bamboo shoots (DBS) and different predictor variables used for modeling DBS for moso bamboo (number of bamboo shoots (NBS), mean height to crown base (MHCB), hydrolytic nitrogen (HN), and available potassium (AK)).

In practice, the matrix form of the NLME is as follows.

$$
\begin{cases}
y_i = f(\theta_i, x_i) + \xi_i \\
\theta_i = A_i b + B_i \mu_i \\
\mu_i \sim N(0, D); \xi_i \sim N(0, R_i)
\end{cases}
\tag{8}
$$

where $y_i$ represents the observed value of the DBS, $x_i$ is a vector for the observed predictor variables on the $i$th block, $x_i$ is the design matrix corresponding to the non-random effect parameter b, and $B_i$ is the design matrix of the random parameter $\mu_i$. We assumed that $\mu_i$ follows a normal distribution with a mean of zero and variance of $D$, which is given by

$$
D = \begin{bmatrix}
\sigma^2_{\mu_{i1}} & \sigma_{\mu_{i1}\mu_{i2}} \\
\sigma_{\mu_{i2}\mu_{i1}} & \sigma^2_{\mu_{i2}}
\end{bmatrix}
\tag{9}
$$

Hypotheses $\mu_i$ and $\xi_i$ are independent. The residual vector is defined by $\xi_i \sim N(0, R_i)$, where the variance–covariance matrix ($R_i$) is calculated using the following formula:

$$
R_i = \sigma^2 G_i^{0.5} \Gamma_i G_i^{0.5}
\tag{10}
$$

where $\sigma^2$ is the residual variance common to all blocks, $\mathbf{G}_i$ is a diagonal matrix related to heteroscedasticity within the block, and $\mathbf{\Gamma}_i$ is a matrix used to explain the within-block autocorrelation structure of the residuals. As our residual analysis showed insignificant autocorrelations of the observations within the same subject (block), we reduced the $\mathbf{\Gamma}_i$ to identity matrices.

Three common variable stability functions (the exponential function, power function, and constant plus power function (Equations (11)–(13)) were applied to explain heteroscedasticity. After evaluating the performance of each function, the one with the best performance was selected according to the AIC value.

$$Var(\xi_{ij}) = \sigma^2 \exp(2\gamma NBS_{ij}) \tag{11}$$

$$Var(\xi_{ij}) = \sigma^2 NBS_{ij}^{2\gamma} \tag{12}$$

$$Var(\xi_{ij}) = \sigma^2 (\gamma_1 + NBS_{ij}^{2\gamma_2})^2 \tag{13}$$

### 2.6. Model Evaluation

The use of an independent dataset to evaluate the DBS model was more effective. However, due to the limited amount of data, we were unable to implement this in our study. Instead, we used leave-one-out cross-validation (LOOCV), a commonly used method, to evaluate the model [23,24]. The data were grouped according to the sample plots, and only one sample plot was reserved in each complete data set. The data set with one sample plot removed was used to fit the model, and the resulting model was used to predict the number of *DBS* in the deleted sample plot. We repeated this 64 times. The difference between the predicted *DBS* value and the observed *DBS* value was used to calculate $R^2$, *TRE*, and *RMSE*, which are defined below (Equations (14)–(16)).

$$R^2 = 1 - \frac{\sum_{i=1}^n (DBS_i - \hat{DBS}_i)^2}{\sum_{i=1}^n (DBS_i - \overline{DBS})^2} \tag{14}$$

$$TRE = \sum_{i=1}^n \left| DBS_i - DBS_i \right| \Big/ \sum_{i=1}^n DBS_i \tag{15}$$

$$RMSE = \sqrt{\frac{1}{n} \sum_{i=1}^n (DBS_i - DBS_i)^2} \tag{16}$$

where $n$ is the number of sample plots, $DBS_i$ is the actual value of degraded bamboo shoots in the *i*th sample plot, $\hat{DBS}_i$ is the estimated value of degraded bamboo shoots in the *i*th sample plot, and $\overline{DBS}$ is the average observed value of degraded bamboo shoots.

We estimated the models using the glmmTMB package [34] and the nlme package in R 3.6.3 [35].

## 3. Results

### 3.1. Basic Models

Because the ZINB and HNB models did not converge, these were not considered in this study. Model parameter estimations (Equations (1)–(3), (5), and (7)) were significantly different from 0, except for $\beta_0$ and $\beta_3$ in the logistic model, and $\beta_2$, $\beta_3$, and $\beta_4$ in the NB model ($p < 0.05$; Table 4). The NB, ZIP, and HP models exhibited identical fit statistics,

which were poorer than that of the PS model. DBS was positively correlated with NBS and HN but negatively correlated with MHCB and AK. Compared with the other models, the logistic model had better fitting statistics, the largest R², and the smallest RMSE and TRE values.

**Table 4.** Parameter estimates and fit statistics of seven basic models (Equations (1)–(3), (5), and (7)).

| Parameter | | Poisson (M1) | NB (M2) | ZIP (M3) | HP (M5) | Logistic (M7) |
|---|---|---|---|---|---|---|
| | a | | | | | 1676 *** |
| | | | | | | (216.3000) |
| Zero component | $\alpha_0$ | | | 8.6077 * | 8.6085 * | |
| | | | | (4.0411) | (4.0412) | |
| | $\alpha_1$ | | | −0.1843 * | −0.1843 * | |
| | | | | (0.0731) | (0.0732) | |
| Count component | $\beta_0$ | 6.7810 *** | 5.4638 ** | 7.6000 *** | 7.6000 *** | 0.3075 |
| | | (0.0670) | (1.8701) | (0.0644) | (0.0644) | (3.1060) |
| | $\beta_1$ | 0.0005 *** | 0.0006 ** | 0.0004 *** | 0.0004 *** | −0.0018 *** |
| | | ($5.81 \times 10^{-6}$) | (0.0002) | ($6.21 \times 10^{-6}$) | ($6.21 \times 10^{-6}$) | ($4.87 \times 10^{-4}$) |
| | $\beta_2$ | −0.2819 *** | −0.1519 | −0.0076 *** | −0.0076 *** | 0.1417 ** |
| | | (0.0010) | (0.3404) | ($1.06 \times 10^{-2}$) | ($1.06 \times 10^{-2}$) | ($5.03 \times 10^{-2}$) |
| | $\beta_3$ | −0.0038 *** | −0.0226 | −0.0549 *** | −0.0548 *** | 0.7761 |
| | | (0.0008) | (0.0222) | (0.0008) | (0.0008) | (0.5046) |
| | $\beta_4$ | 0.0103 *** | 0.0080 | 0.0009 *** | 0.0008 *** | −0.0427 ** |
| | | (0.0002) | (0.0073) | (0.0002) | (0.0002) | (0.0149) |
| R² | | 0.5268 | 0.4691 | 0.4861 | 0.4861 | 0.6022 |
| RMSE | | 429.9827 | 455.4711 | 448.0134 | 448.0134 | 394.1936 |
| TRE | | 30.3217 | 32.2449 | 28.7537 | 28.7537 | 25.3112 |

Note: *** $p < 0.0001$, ** $p < 0.001$, * $p < 0.05$. Values in parentheses are standard errors. NB is the negative binomial model, ZIP is the zero-inflated Poisson, and HP is the hurdle Poisson. M is model.

We evaluated the simulation effects of predictors (MHCB) and soil nutrient content factors (AK and HN) describing the characteristics of bamboo forests on the DBS of the logistic model (Figure 5). The three predictors have a significant impact on the change in DBS (Figure 5). DBS decreased with increasing MHCB and AK but decreased with increasing HN. AK and HN contributed the most to variations in DBS, followed by MHCB.

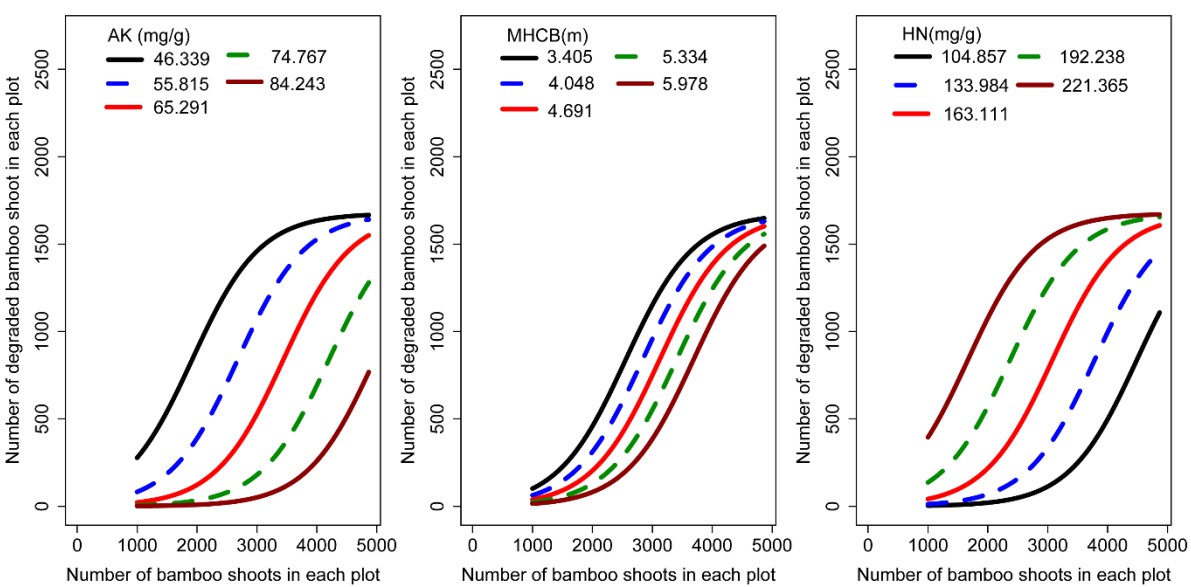

**Figure 5.** Effects of MHCB, AK, and HN on the DBS. The curves were produced using parameter estimates in Table 3 (logistic model).

*3.2. NLME Models*

The estimated parameters and fit statistics for Models (17)–(21) are presented in Table 5. The parameter estimates of the models (Equations (1)–(3), (5), and (7)) were significantly different from 0, except for $\beta_0$ and $\beta_3$ in M21, and $\beta_2$, $\beta_3$, and $\beta_4$ in M18 ($p < 0.05$; Table 5). Compared with Models (1)–(3),(5),(7), except for M18, the TRE and RMSE of Models (17),(19)–(21) significantly decreased, and $R^2$ increased, respectively. Model (21) had the smallest RMSE and TRE, and the highest $R^2$. Therefore, we used the logistic NLME model for further analysis.

**Table 5.** Parameter estimates and fit statistics of the one-level nonlinear mixed-effects DBS models (Equations (1)–(3), (5), and (7)).

| Parameter | | Poisson (M17) | NB (M18) | ZIP (M19) | HP (M20) | Logistic (M21) |
|---|---|---|---|---|---|---|
| | a | | | | | 1897.1424 *** (275.5338) |
| Zero component | $\alpha_0$ | | | 8.6077 * (4.0361) | 8.6080 * (4.0126) | |
| | $\alpha_1$ | | | −0.1843 * (0.0731) | −0.1944 * (0.0733) | |
| Count component | $\beta_0$ | 4.7960 *** (0.2171) | 5.4638 ** (1.8701) | 5.3030 *** (0.2214) | 5.3027 *** (0.2208) | 1.4120 (2.4835) |
| | $\beta_1$ | 0.0007 *** ($8.166 \times 10^{-6}$) | 0.0006 ** (0.0002) | 0.0005 *** ($8.512 \times 10^{-6}$) | 0.0005 *** ($8.412 \times 10^{-6}$) | −0.0015 *** (0.0004) |
| | $\beta_2$ | −0.2045 *** (0.0125) | −0.1519 (0.3404) | −0.0311 * (0.0134) | −0.0299 * (0.0130) | 0.1266 *** (0.0362) |
| | $\beta_3$ | −0.0538 *** ($8.984 \times 10^{-4}$) | −0.0226 (0.0222) | −0.0673 *** ($8.928 \times 10^{-4}$) | −0.0655 *** ($8.902 \times 10^{-4}$) | 0.5535 (0.4239) |
| | $\beta_4$ | 0.0226 *** ($4.153 \times 10^{-4}$) | 0.0080 (0.0073) | 0.0233 *** ($4.434 \times 10^{-4}$) | 0.0212 *** ($4.404 \times 10^{-4}$) | −0.0407 *** (0.0118) |
| Covariance matrix of random effects variance | block | 0.1821 | $4.414 \times 10^{-9}$ | 0.1924 | 0.1920 | 0.1107 |
| $R^2$ | | 0.6183 | 0.4690 | 0.5991 | 0.5990 | 0.6596 |
| RMSE | | 386.2009 | 455.4711 | 395.6937 | 394.6932 | 364.7028 |
| TRE | | 23.9796 | 32.2448 | 22.2586 | 22.2183 | 20.8995 |

Note: *** $p < 0.0001$, ** $p < 0.001$, * $p < 0.05$. The value (0.1107) is generated when the random effect is added to the coefficient of variable AK of the logistic model; other random effects were added to the intercept. All other parameters, symbols, and definitions are the same as those listed in Table 3.

Among the three tested variance functions (Equation (14)–(16)), the power function (Equation (15)) most effectively accounted for the variance heteroscedasticity (Table 6, Figure 6). The RMSE and TRE values of the NLME DBS Model (21) were (RMSE = 364.7028; TRE = 20.8995) based on the full dataset. Therefore, we added the random effect to $\beta_3$, which produced the largest $R^2$ (0.6596) and was assumed to be an optimal NLME model to estimate DBS, that is, the final NLME model form given by

$$DBS_{ij} = \frac{a}{1 + e^{[\beta_0 + \beta_1 NBS_{ij} + \beta_2 MHCB_{ijk} + (\beta_3 + \mu) AK_{ijk} + \beta_4 HN_{ij}]}} + \xi_{ij}$$

where $\mu$ is the random effect at block level for $\beta_3$. $\xi_{ij} \sim N(0, R_i = \sigma^2 G_i^{0.5} \Gamma_i G_i^{0.5}), G_i = diag(\sigma^2 NBS_{i1}^{2\gamma}, ... \sigma^2 NBS_{in}^{2\gamma}), \Gamma_i = I_i$. The other parameters and variables were the same as described above.

**Table 6.** Comparisons among three variance functions (exponential function, power function, and constant plus power function) of the NLME DBS model (LL, log-likelihood).

| Variance functions | NLME DBS Model | |
| --- | --- | --- |
| | AIC | LL |
| Equation (11) | 949.0738 | −466.5369 |
| Equation (12) | 948.1784 | −465.0892 |
| Equation (13) | 950.1784 | −465.0892 |

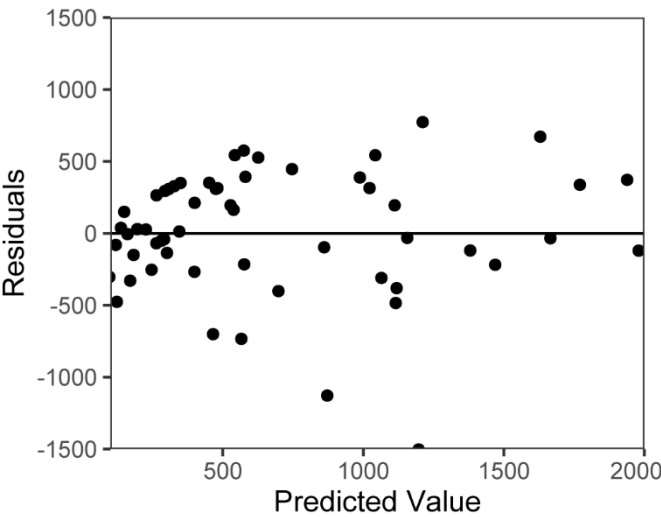

**Figure 6.** Residual distribution of Model (21).

*3.3. Parameter Estimates*

Model parameter estimations were significantly different from 0, except for $\beta_3$, in the NLME DBS Model (21). After introducing the parameter estimates, the NLME DBS Model (21) is:

$$DBS_{ij} = \frac{2146.0780}{1 + e^{[1.8096 - 0.0012 NBS_{ij} + 0.0677 MHCB_{ijk} + (0.3627 + \mu_0) AK_{ijk} - 0.0210 HN_{ij}]}} + \xi_{ij}$$

where

$$\mu_i = [\mu_0] : N\left\{[0], \psi = (1.13 * 10^{-10})\right\}$$
$$\xi_{ij} \sim N(0, R_i = 137382.4 G_i^{0.5} \Gamma_i G_i^{0.5})$$

$$G_i = diag(0.0092NBS_{i1}^{1.0635}, \ldots 0.0092NBS_{in}^{1.0635})$$

$$\Gamma_i = I_i$$

Using the variance function can reduce the variance in the Model (27) to a certain extent, which indicates that a large part of the DBS change is explained by the block-level random effects.

*3.4. Model Evaluation*

We evaluated the prediction performance of the logistic NLME model, which showed best-fit statistics using the LOOCV method. The validation showed a large proportion of variation in DBS ($R^2$ = 0.4324, RMSE = 467.9945, and TRE = 57.4346). This result further shows that blocks have a large random impact on DBS, and the prediction ability of NLME DBS Model (21) at the block level was the best.

## 4. Discussion

We evaluated seven functions commonly used in modeling count data [23,24] and attempted to evaluate other model forms, such as the zero-inflated negative binomial and hurdle negative binomial models; however, they were not retained for subsequent analyses because of their non-convergence. Collinearity between the predictor variables was controlled with VIF, which is a common practice, while multiple predictor variables were involved in fitting the models. Among the various potential predictor variables that we evaluated, only four, namely, NBS, MHCB, HN, and AK, showed significant contributions to the model.

Other studies have also used NBS and NB to predict DBS [16,36], as these variables could have a remarkably strong influence on DBS variations across bamboo forests. NBS and NB may reflect the nutrient supply in the stand and the impact of the site quality of the bamboo forest [5,18]. Zhang et al. (2012) used BA in their counting model (mortality model) for *Larix olgensis*, which reflects stand DBH growth and may describe DBS caused by stand competition [37–39]. The random effect could account for the variability across forest blocks, which could be caused by the effects of site quality and other environmental factors, including climate. The combined effects of these factors on variations in DBS can be described by introducing a random effect into the model.

In the present study, the effect of NBS was positive (Tables 4 and 5). The results of this study are consistent with those of Liao and Huang (1984) and Xu et al. (2008). Under certain growth conditions, NBS resulted in more degraded bamboo. This may be because the nutrient content in the sample plot was limited and could only meet the survival requirements of certain bamboo shoots.

In contrast, MHCB negatively correlated with DBS. HCB reflects the growth, vitality, and productivity of bamboo [40], as well as the level of competition within the bamboo forest stand [41,42]. The smaller the MHCB, the larger the crown and the faster the nutrient metabolism, indicating that the bamboo competitiveness of the original bamboo forest is stronger, and the number of DBS has increased.

DBS were significantly negatively correlated with AK (Tables 4 and 5, Figure 5). Enzyme activation is one of the most important functions of AK in plant growth. AK is closely related to many metabolic processes in plants, such as photosynthesis, respiration, and the synthesis of carbohydrates, fats, and proteins. These processes are essential during the early stages of shoot emergence. In the absence of K, the number of DBS can increase. It has been found that AK concentrations in each organ of the growing shoots were higher in the younger parts than in mature organs [43]. More AK is required during the shooting period. Therefore, AK content is a key factor in determining the number of DBS. We also observed a negative correlation between DBS and HN. The utilization of nitrogen was mainly concentrated after the leaf development of new bamboo [44] because, at this

stage, the physiological metabolism of moso bamboo is vigorous, and dry matter accumulates rapidly.

Some studies have reported that other stand variables, such as arithmetic mean diameter and stand density, contribute significantly to DBS prediction [18,23,24,45]. We also evaluated these variables, but the results were not statistically significant. Both stand arithmetic mean diameter and stand density were derived from DBH and thus showed little effect on the DBS model.

Stand measures, such as BA and quadratic mean DBH (QMD), could be obtained more accurately than other measures, which are the most commonly used predictors in any forestry model [37–39]. However, our model showed non-significant effects for these variables in the DBS model. This may be due to the small DBH differences among the bamboo in the study area, where the density of bamboo forests fluctuates within a limited range. However, the stand variables NBS and NB may have a significant effect on DBS. The NBS was positively correlated with DBS, indicating that increased NBS also promoted DBS (Figure 5). Because soil nutrients are limited, they can only supply a certain amount of bamboo growth. When NBS increased, the number of DBS also increased. Some studies (e.g., [46,47]) have shown that fertilization and reclamation can reduce DBS to a certain extent by supplementing certain soil-available nutrients.

As an important index of stand characteristics, DBS determine the growth and harvest of bamboo forests [5]. This is because bamboo grows from bamboo shoots. For cultivated bamboo, DBS consume much of the nutrients from the mother bamboo, but this does not increase yield. Therefore, studies focused on modeling DBS are of great importance for bamboo forest management.

The hierarchical data problem was solved by introducing block-level random effects in the $\beta_3$ parameter (e.g., the correlation of observed values of different plots and clumps in the same block). Previous studies on forest mortality (data type was also counting form) also found that the mixed-effects model was better than the traditional nonlinear least-squares model [4,23,24]. When the random effect (block) was included in the logistic model, the prediction accuracy of LOOCV improved to a certain extent, indicating that the block had an impact on degraded bamboo shoots. This also largely justifies the application of mixed-effects modeling in our study.

During the modeling process, by evaluating the impact of individual bamboo, stands, and soil nutrient content variables on the number of DBS, variables (MHCB: mean height to crown base, HN: hydrolytic nitrogen, AP: available potassium) that have a greater impact on the number of DBS were selected, and these variables were adjusted through some forest management, such as applying N and P fertilizer in the growing season, or carrying out tending measures (pruning) in the bamboo forest to provide sufficient growth space and nutrients for new bamboo, to reduce the number of DBS, increase the biomass of bamboo forests, and maintain the stability of bamboo ecosystem. It can provide an important basis for investigating the carbon sink properties of bamboo forests and help formulate more effective bamboo forest management plans.

In this study, we only considered the number of DBS, and not the depth of shoots or topographic factors, such as altitude, slope, and aspect of the sample plot. Some studies have pointed out that changes in nutrient levels are also key factors affecting the survival of bamboo shoots [5,48]. Thus, nutrient changes during DBS might also be important for understanding nutrient deficiencies and how to supplement the soil [49–51]. Therefore, it is important to study the nutritional changes associated with DBS.

## 5. Conclusions

We established a nonlinear mixed-effects model to appropriately describe the variations in DBS of moso bamboo. The stand and soil content variables used as predictors, such as NBS, MHCB, HN, and AK, were identified as the major factors affecting DBS. Introducing the block-level random effect improves the fitting effect of the model. Among the various model formulations (basic and mixed models), random effects for the logistic

model described the largest variation in DBS. Additionally, we found that DBS in bamboo stands increased with decreasing MHCB and AK but decreased with decreasing NBS and HN. Reducing the number of DBS and increasing biomass through fertilization, tending, and other measures can provide an important basis for investigating the carbon sink properties of bamboo forests and help formulate more effective bamboo forest management plans.

**Author Contributions:** X.Z. (Xiao Zhou), X.Z. (Xuan Zhang), S.F., Z.Y., C.L., Y.Z., and F.G. collected the data; X.Z. (Xiao Zhou), X.Z. (Xuan Zhang), C.L., Z.Y., S.F., Y.Z., and F.G. analyzed data; X.Z (Xiao Zhou), and F.G. wrote the manuscript, contributed critically to improve the manuscript, and gave final approval for publication. All authors have read and agreed to the published version of the manuscript.

**Funding:** This research was supported by the Basic Scientific Research Fund of the International Center for Bamboo and Rattan (grant no. 1632021003).

**Data Availability Statement:** Not applicable.

**Acknowledgments:** We would like to thank the basic scientific research funding of the International Center for Bamboo and Rattan (grant no. 1632021003) for the financial support provided for this study. We are thankful to the two anonymous reviewers for their insightful comments and suggestions, which helped improve the manuscript.

**Conflicts of Interest:** The authors declare no conflict of interest.

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
