# Peer review of "Modeling Degraded Bamboo Shoots in Southeast China"

_forests, doi:10.3390/f13091482_

Round 1

Reviewer 1 Report

This paper presents a prediction of the change in degraded bamboo shoots using simulation with mixed effects. The findings from the study are interesting, relevant and have the potential for practical application in bamboo forest management. Several questions and comments need to be clarified.

1. At the edge of the range of any plant species, habitat is far worse than optimal living conditions (otherwise the plant would have inhabited the territory, and the boundary of the range was more distant). Why were the authors researching the growth of bamboo at the edge of the range (“the northeast edge of the moso bamboo distribution area”)?

2. Why were only soil conditions studied, and climate factors and pest/disease effects not included in the modelling?

3. “The investigation time was from March to May 2019. … Soil samples were collected from 64 sample plots in May 2018”.

Please clarify the timing of all of this research. Soil studies (May 2018) were conducted before the main experiment (March to May 2019)?

4. “Studies have found that the nutrient absorbing fine roots of moso bamboo in this area are mainly distributed in the 0–10-cm soil layer [30]. Therefore, this study mainly studied the contribution of soil nutrients in the 0–10-cm layer.”

The study of fine roots, to which the authors refer, was conducted in October 2020 (ref 30. https://doi.org/10.3390/f12060793). How did the authors determine for the year of the experiment (2019), or even for two years (if the soil sampling date is correct - May 2018), that there is a need to take samples at this depth (based on 2020 data)?

Author Response

Comments and Suggestions for Authors

This paper presents a prediction of the change in degraded bamboo shoots using simulation with mixed effects. The findings from the study are interesting, relevant and have the potential for practical application in bamboo forest management. Several questions and comments need to be clarified.

  1. At the edge of the range of any plant species, habitat is far worse than optimal living conditions (otherwise the plant would have inhabited the territory, and the boundary of the range was more distant). Why were the authors researching the growth of bamboo at the edge of the range (“the northeast edge of the moso bamboo distribution area”)?

Response: thanks for your advice, we modified this in the manuscript.

Although the study area is on the northeast edge, bamboo can also grow normally in this area. Because of the importance of bamboo growth in this area and the urgency of business measures for the study of this area, the growth of bamboo forests in this area is studied. And the study of this region can also provide theoretical basis and technical methods for the study of other regions.

See line 30-35, 91-103.

  1. Why were only soil conditions studied, and climate factors and pest/disease effects not included in the modelling?

Response: thanks for your advice, we modified this in the manuscript.

See line 53-58.

In bamboo forests, bamboo shoots can be divided into normal and abnormal DBS. Normal DBS are mainly caused by the competition of water, nutrients, and light conditions in new bamboo. Abnormal DBS are mainly caused by man-made random interference or natural disasters, such as extreme weather, forest fires, snow disasters, droughts, forest diseases, and insect pests [4,5]. Owing to the randomness of abnormal DBS, only normal DBS stands are typically studied

  1. “The investigation time was from March to May 2019. … Soil samples were collected from 64 sample plots in May 2018”.

Please clarify the timing of all of this research. Soil studies (May 2018) were conducted before the main experiment (March to May 2019)?

Response: thanks for your advice, we modified this in the manuscript.

See line 158.

I am sorry for this problem due to my writing problem. The soil collection time is March 2019. When designing the sample plot, the soil has been sampled.

  1. “Studies have found that the nutrient absorbing fine roots of moso bamboo in this area are mainly distributed in the 0–10-cm soil layer [30]. Therefore, this study mainly studied the contribution of soil nutrients in the 0–10-cm layer.”

The study of fine roots, to which the authors refer, was conducted in October 2020 (ref 30. https://doi.org/10.3390/f12060793). How did the authors determine for the year of the experiment (2019), or even for two years (if the soil sampling date is correct - May 2018), that there is a need to take samples at this depth (based on 2020 data)?

Response: thanks for your advice, we modified this in the manuscript.

See line 154-157.

In the experiment, we took samples of 0-10, 10-20, 20-30 and 30-50cm soil layers. Because most studies have found that soil nutrients are episodic, and some studies have found that the roots of bamboo forests are mainly distributed at 0-60cm, and mainly at 0-10cm, and my research position is the same as that of the researchers in this document, and the site conditions are similar.

Reviewer 2 Report

The article is very relevant and interesting, dedicated to modeling degraded bamboo shoots. I believe that the work will make a great contribution not only to forest science, but should also be used in forest management in subtropical and tropical zones. There are a few minor and easily fixed remarks:

1. In Figure 1, it is not clear where the key areas of the study are located. It makes sense to put a coordinate grid on maps. It is possible to show the localization on the general map of China.

2. The conclusions are very brief. It would be nice to expand them.

Author Response

Comments and Suggestions for Authors

The article is very relevant and interesting, dedicated to modeling degraded bamboo shoots. I believe that the work will make a great contribution not only to forest science, but should also be used in forest management in subtropical and tropical zones. There are a few minor and easily fixed remarks:

  1. In Figure 1, it is not clear where the key areas of the study are located. It makes sense to put a coordinate grid on maps. It is possible to show the localization on the general map of China.

Response: thanks for your advice. We modified this in the manuscript.

See line 124-125.

  1. The conclusions are very brief. It would be nice to expand them.

Response:  thanks for your advice. We modified this in the manuscript.

See line 525-534.

Reviewer 3 Report

The manuscript under review focuses on an interesting topic, as is the aim to modelling the number of degraded bamboo shoots as a function of different stand and soil variables. While the general aim is a worthwhile topic for research, and the available data seems quite valuable as for dealing with the objective, I have detected several methodological flaws which prevent me from recommending publication. In the next paragraphs I’ll expose, section by section, these main concerns, together with other minor errors, as well as some suggestions for improving.

INTRODUCTION

-          Page 1. 1st paragraph: as many of the potential readers of Forests journal may be non-familiar with bamboo forests and diseases, I strongly recommend to include here a more in Deep description of the DBS process, and how is it affecting bamboo plantations.

-          Page 1. 3rd paragraph. While in the previous paragraph the authors indicate tht DBS may be casued by 4 factors (nutrition, diseases, climate and bamboo forest conditions) here they reduce the potential causes to only two: nutrition and stand variables, justifying that the other causes are uncommon (what is a contradiction with the previous paragraph). I encourage the authors to either give  a more detailed and sound justification on  why only select these two groups of potential explanatory covariate.

-          Page 2. Last paragraph of the section. I miss some sound hypothesis to be contrasted  based on previous studies

MATERIAL and MEHODS

Major comments

Data collection

A lot of information is missed on data acquisition process. It is not clear how plots were selected within the forest (systematically, subjective). Plot general characteristics (size, shape) are also missed. While in the methods section the authors claim for including a block random effect, nothing about blocking has been presented in material section. Were the plots selected within blocks? Which was the cause for this blocking?

Concerning measurements, nothing is said on how number of bamboos shoot, new shoots and degraded shots was assessed. Was it by means of direct counting? Subsampling within the plot (i.e. quadrats)?

In table 1 units and spatial scale (are these data referred to the plot or the hectare) should be provided

Finally, figure 2 does not show the distribution patterns of DBS, and the title of the axis may be worong. Ordinates Y axis does not show the frequency of DBS (this would be the number of plots with a given value of DBS) but the observed raw values of DBS in the plots. On the other hand, abscissa X axis is not showing the DBS /ha (between o and approx. 60). In my opinion figure 2 show an ordination of the 64 plots showing the real value of DBS per ha. In addition, the number of DBS seem to be presented on discrete steps of 100 (0 – 100 – 200… up to 2700). Do the author count all the DBS in a plot, or do they make an estimate in hundreds?

Covariate selection

My first methodological main criticism is related with the process of covariate selection, that in my opinion is a complete nonsense. The authors have prioritized preventing multicollinearity over other much more logical issues, as could be the rate of variability on the response variabl explained by the predictor. Multicollinearity has sense in the presence of other variables, thus a single covariate has no multicollinearity at all. By pooling all the covariates and eliminating those with WIF > 2 (a real severe restriction) the authors may be missing a lot of useful information. It seems a much more logical way for model construction to first carry out a preliminary exploratory correlation analysis between DBS and potential covariates, and between the potential covariates among them. Based on this analysis the try to construct a model using some kind of sequential procedure, and finally, check for if some of the retained covariates show multicollinearity.

Candidate models

The authors should justify why they have selected two models (ZIP and Hurdle) commonly used when an excess of zeroes is detected, when according to figre 2 they do not have excess of zeroes at all (may be 3 or 4 plots in 64?). On the other hand, why other flexibla models for count data that can easily deal with a few zeroes (mainly negative binomial) have been ignored. In discussion the authors say this model was tested, but no results have been shown (and given its large flexibility it is very strange that ZIP converge and NB does not).

Concerning the “logistic” model, why the authors have decided not to expand the asymptote “a” of the model over different predictors?. On its current form predictions are censored to a maximum value given by “a”, irrespective of the stand / nutrition conditions, which seem quite illogical.

Mixed effect models

The authors should describe how blocking of the plots was carried out. Moreover, even if existing a sound reason for this blocking, it is not logical at all that the authors use it for justifying the inclusion of a randomblock effect on the logistical model, but not on the other candidate models. Inherent correlation due to the blocks may be affecting the estimation of the standard error of the parameters for all the candidate models, thus a mixed modelling approach will be required for the Poisson, ZIP, hurdle and logistic model.

Page 8, first line: “due to the nonlinear patterns of data”. Figure 3 may evidence many things except nonlinear relations (e.g MCHB vs DBS show a clear lack of relation, and NBS vs DBS is clearly linear one)

Eq 9. The covariance terms should not include the square term

Model evaluation

It is not clear if the statistics here presented were used in the LOOCV process. If so, how werethey estimated (average value of the R2 computed on each realization of the cross validation)?

RESULTS

-          Table 2. By definition it is not possible that ZIP and Hurdle Poisson lead to the same results and statistics. I suggest the authors to check their programming statements

-          Section 3.2. As previously stated, it is not justified at all why include random effects in the logistic model and not include them in the other tested models

Author Response

Comments and Suggestions for Authors

The manuscript under review focuses on an interesting topic, as is the aim to modelling the number of degraded bamboo shoots as a function of different stand and soil variables. While the general aim is a worthwhile topic for research, and the available data seems quite valuable as for dealing with the objective, I have detected several methodological flaws which prevent me from recommending publication. In the next paragraphs I’ll expose, section by section, these main concerns, together with other minor errors, as well as some suggestions for improving.

INTRODUCTION

-          Page 1. 1st paragraph: as many of the potential readers of Forests journal may be non-familiar with bamboo forests and diseases, I strongly recommend to include here a more in Deep description of the DBS process, and how is it affecting bamboo plantations.

Response: thanks for your advice. We modified this in the manuscript.

See line 30-35

DBS is a phenomenon in which the development of new bamboo stops after the shoots are unearthed because of insufficient nutrition in the soil, foreign pests and diseases, sudden cold or dry weather, competition in the bamboo forest, such as the number of bamboo shoots (NBS), and bamboo forest structure, and they then die and cannot fur-ther develop into bamboo.

-          Page 1. 3rd paragraph. While in the previous paragraph the authors indicate tht DBS may be casued by 4 factors (nutrition, diseases, climate and bamboo forest conditions) here they reduce the potential causes to only two: nutrition and stand variables, justifying that the other causes are uncommon (what is a contradiction with the previous paragraph). I encourage the authors to either give  a more detailed and sound justification on  why only select these two groups of potential explanatory covariate.

Response: thanks for your advice. We modified this in the manuscript.

See line 36-44, 45-58

-          Page 2. Last paragraph of the section. I miss some sound hypothesis to be contrasted  based on previous studies

Response: thanks for your advice. We modified this in the manuscript.

See line 91-94.

MATERIAL and MEHODS

Major comments

Data collection

A lot of information is missed on data acquisition process. It is not clear how plots were selected within the forest (systematically, subjective). Plot general characteristics (size, shape) are also missed. While in the methods section the authors claim for including a block random effect, nothing about blocking has been presented in material section. Were the plots selected within blocks? Which was the cause for this blocking?

Response: thanks for your advice. We modified this in the manuscript.

See line 117-121.

Concerning measurements, nothing is said on how number of bamboos shoot, new shoots and degraded shots was assessed. Was it by means of direct counting? Subsampling within the plot (i.e. quadrats)?

Response: thanks for your advice. We modified this in the manuscript.

See line 149.

In table 1 units and spatial scale (are these data referred to the plot or the hectare) should be provided

Response: thanks for your advice. We modified this in the manuscript.

See line 185-186.

Finally, figure 2 does not show the distribution patterns of DBS, and the title of the axis may be worong. Ordinates Y axis does not show the frequency of DBS (this would be the number of plots with a given value of DBS) but the observed raw values of DBS in the plots. On the other hand, abscissa X axis is not showing the DBS /ha (between o and approx. 60). In my opinion figure 2 show an ordination of the 64 plots showing the real value of DBS per ha. In addition, the number of DBS seem to be presented on discrete steps of 100 (0 – 100 – 200… up to 2700). Do the author count all the DBS in a plot, or do they make an estimate in hundreds?

Response: thanks for your advice. We modified this in the manuscript.

See line 185-186.

This figure shows the DBS in each sample plot. The vertical axis shows the Dbs in each plot / ha. The abscissa represents the plot number.

Covariate selection

My first methodological main criticism is related with the process of covariate selection, that in my opinion is a complete nonsense. The authors have prioritized preventing multicollinearity over other much more logical issues, as could be the rate of variability on the response variabl explained by the predictor. Multicollinearity has sense in the presence of other variables, thus a single covariate has no multicollinearity at all. By pooling all the covariates and eliminating those with WIF > 2 (a real severe restriction) the authors may be missing a lot of useful information. It seems a much more logical way for model construction to first carry out a preliminary exploratory correlation analysis between DBS and potential covariates, and between the potential covariates among them. Based on this analysis the try to construct a model using some kind of sequential procedure, and finally, check for if some of the retained covariates show multicollinearity.

Response: thanks for your advice. We modified this in the manuscript.

See line 189-208.

Candidate models

The authors should justify why they have selected two models (ZIP and Hurdle) commonly used when an excess of zeroes is detected, when according to figre 2 they do not have excess of zeroes at all (may be 3 or 4 plots in 64?). On the other hand, why other flexibla models for count data that can easily deal with a few zeroes (mainly negative binomial) have been ignored. In discussion the authors say this model was tested, but no results have been shown (and given its large flexibility it is very strange that ZIP converge and NB does not).

Response:thanks for your advice. We modified this in the manuscript.

See line 213-216, 241-249, 269-278, 293-301

We added other counting models and performed calculations

Concerning the “logistic” model, why the authors have decided not to expand the asymptote “a” of the model over different predictors?. On its current form predictions are censored to a maximum value given by “a”, irrespective of the stand / nutrition conditions, which seem quite illogical.

Response:thanks for your advice. We modified this in the manuscript.

I think this model is reasonable. No matter what the nutrient and stand status are, they all have a maximum value, that is, the horizontal asymptote a.

And this model can solve this problem (that is, when soil nutrients are insufficient and competition is large, the number of bamboo shoots retreated is the largest, that is, DBS is close to the maximum value a.

Mixed effect models

The authors should describe how blocking of the plots was carried out. Moreover, even if existing a sound reason for this blocking, it is not logical at all that the authors use it for justifying the inclusion of a randomblock effect on the logistical model, but not on the other candidate models. Inherent correlation due to the blocks may be affecting the estimation of the standard error of the parameters for all the candidate models, thus a mixed modelling approach will be required for the Poisson, ZIP, hurdle and logistic model.

Response:thanks for your advice. We modified this in the manuscript.

We added the basis for block division. We select the model with better fitting effect to develop the mixed effect model by comparing the fitting quality of the model.

See line 400-411.

Page 8, first line: “due to the nonlinear patterns of data”. Figure 3 may evidence many things except nonlinear relations (e.g MCHB vs DBS show a clear lack of relation, and NBS vs DBS is clearly linear one)

Response:thanks for your advice. We modified this in the manuscript.

See line 317-323.

Eq 9. The covariance terms should not include the square term

Response:thanks for your advice. We modified this in the manuscript.

See line 336.

Model evaluation

It is not clear if the statistics here presented were used in the LOOCV process. If so, how werethey estimated (average value of the R2 computed on each realization of the cross validation)?

Response:thanks for your advice. We modified this in the manuscript.

See line 355-365.

RESULTS

-          Table 2. By definition it is not possible that ZIP and Hurdle Poisson lead to the same results and statistics. I suggest the authors to check their programming statements

Response:thanks for your advice. We modified this in the manuscript.

See line 385-386.

-          Section 3.2. As previously stated, it is not justified at all why include random effects in the logistic model and not include them in the other tested models

Response:thanks for your advice. We modified this in the manuscript.

See line 400-410.

Round 2

Reviewer 1 Report

The authors carefully considered all comments and questions and corrected the manuscript. But one issue remains open.

Please amend reference number 30. Replace the "fine roots research" article published in 2019 or earlier. Since the research was carried out in 2019, it is impossible to refer to the methodological features of the research, defined in 2020.

Author Response

thanks for your advice.

We use the following reference instead of the original reference。

  • Fan, SH., Xiao, FM., Wang, SL., et al. 2009. Fine root biomass and turnover in moso bamboo plantation in Huitong forest station, Hunan province. Scientia Silvae sinicae. 45 (07): 1-6. (in chinese)

Reviewer 3 Report

This is my second review on thus manuscript and I must acknowledge that the authors have made an impressive effort in taking into account all the suggestions I made in my previous revission. In its current state I consider that the text has been substantially improved, and now constitues an interesting piece of work focusing on an innovative topic, as is modelling bamboo disease. Due to this I recommend publication. 

Author Response

thanks for your advice。

thanks